# Investigation of the Alternations in Lipid Oxidation and Lipase Activity in Air-Dried Hairtail (*Trichiurus lepturus*) during Chilled Storage

**DOI:** 10.3390/foods13020229

**Published:** 2024-01-11

**Authors:** Yuexiang Zhan, Jiagen Li, Taiyu Li, Kai Xie, Chuanhai Tu, Zhiyu Liu, Jie Pang, Bin Zhang

**Affiliations:** 1Pisa Marine Graduate School, Zhejiang Ocean University, Zhoushan 316022, China; jimmy980526@126.com (Y.Z.); jiagen10161828@outlook.com (J.L.);; 2Key Laboratory of Health Risk Factors for Seafood of Zhejiang Province, College of Food Science and Pharmacy, Zhejiang Ocean University, Zhoushan 316022, China; 3Fisheries Research Institute of Fujian, Xiamen 350025, China; 4College of Food Science, Fujian Agriculture and Forestry University, Fuzhou 350002, China

**Keywords:** hairtail, water content, water activity, lipid oxidation, lipase activity, storage

## Abstract

The effects of water content and water activity on the lipid stability of air-dried hairtail (*Trichiurus haumela*) were investigated during chilled storage. Air-dried hairtail samples with high and low water contents were comparatively analyzed over 8 days of storage at 4 °C. The results indicated that the decreases in water activity and increases in the NaCl content significantly inhibited lipid oxidation in the air-dried hairtail samples. The peroxidation value (PV), conjugated diene value (CD), thiobarbituric acid reactive substance (TBARS) value, and p-anisidine value (p-AnV) of the air-dried hairtail significantly increased with the extension of storage time. The low water content significantly inhibited the activity of neutral and alkaline lipase, in addition to lipoxygenase, and retarded the rapid increases in the non-esterified fatty acid (NEFA) content in the hairtail samples. The correlation analysis results showed that the TBARS, p-AnV, and lipase activity were positively correlated in the air-dried hairtail samples, and the lower water content significantly inhibited the progress of lipid oxidation. This study offers a theoretical framework for the industrial processing and storage of air-dried hairtail products.

## 1. Introduction

Hairtail (*Trichiurus lepturus*) is a mid–bottom-layer migratory fish that is widely distributed throughout the temperate and tropical oceans. It is rich in both polyunsaturated fatty acids and essential amino acids that are beneficial for humans. According to the *China Fishery Statistics Yearbook* [1], hairtail has been the most harvested marine fish in China from 1980 to 2022. The production of hairtail in China was 914,649 tons and 903,498 tons in 2021 and 2022, respectively. Hairtail contains high amounts of lipids, especially polyunsaturated fatty acids (PUFAs), accounting for 30% of the total lipids [2]. These lipids are susceptible to oxidation and hydrolysis during processing and storage [3,4]. To offer a unique flavor to air-dried hairtail, eviscerated hairtail is salted and naturally wind-dried under the sun for 2–5 d, which is referred to as the air-drying process [5]. Air-dried hairtail is widely appreciated by consumers in China and coastal Southeast Asia due to its unique flavor and taste, as well as its firm texture. Compared with fresh hairtail, the lipid oxidation in air-dried hairtail is markedly elevated due to the air-dried process [6]. Although various advanced methods for air-dried fish preservation have emerged, chilled storage at 0–4 °C remains the paramount technique for the preservation of marine products and for maintaining their quality [7,8]. However, lipid oxidation in fish muscle leads to the generation of abundant undesirable intermediate products, including free radicals, reactive oxygen species, and peroxides [9]. In fish products, these substances also cause a rancid odor and poor nutritional value [10].

Lipid hydrolases and lipoxygenases are two important branches of enzymes that metabolize lipids. Depending on their hydrolytic activities at various pH levels, lipid hydrolases can be divided into neutral lipase, acidic lipase, and alkaline lipase. These three lipases are usually found in the pancreas, stomach, and intestine of fish samples [11]. Lipases can catalyze the hydrolysis of triglycerides into diglycerides, monoglycerides, fatty acids, and glycerol [12]. The released fatty acids are prone to oxidation, resulting in the generation of hydroperoxides and secondary oxidation products [13]. Neutral and alkaline lipases epitomize two lipase subgroupings, which manifest their optimal activity under neutral and alkaline conditions, respectively. Alkaline lipases predominantly encompass pancreatic lipase and intestinal alkaline lipase, lipases which are secreted within their respective organs [14,15,16]. Neutral lipases primarily comprise hepatic lipase, lipoprotein lipase, and adipose triglyceride lipase, found specifically within liver [17], muscle, and adipose tissues [18,19]. Multiple lipoxygenases catalyze the oxidation process of unsaturated fatty acids. This catalytic process results in an array of oxidation products, such as hydroperoxides, hydroxyketones, and epoxy compounds [20,21,22,23]. The enzymatic activities are influenced by a variety of factors, including water activity, pH, temperature, and salt concentration in the muscle tissues [11,24,25]. Previous studies have also investigated the differences in quality stability during storage at various temperatures for boiled-dried anchovy at a 15% water content [26], as well as the lipid damage from salting in salt-dried, processed *Decapterus maruadsi* and *Trichiurus lepturus* [2].

There have been few studies observing the progression of lipid deterioration in air-dried hairtail products during chilled storage, especially for the variations in lipase activity in the current study. Lipid deterioration during the chilled storage of air-dried hairtail is evaluated under two kinds of initial water content conditions. By exploring the lipid physicochemical properties, this study aims to provide theoretical support for the implementation of industrial-scale production and storage of air-dried hairtail and other dried muscle products.

## 2. Materials and Methods

### 2.1. Chemical Reagents

Chloroform, petroleum ether, phenolphthalein, sodium thiosulfate, thiobarbituric acid (TBA), and triphenylphosphine were supplied by Sinopharm Chemical Reagent Co., Ltd. (Shanghai, China). P-anisidine, dithiothreitol (DTT), Ethylenediaminetetraacetic acid disodium salt (EDTA-2Na), trichloroacetic acid (TCA), p-nitrophenyl laurate (p-NPL), 4-methylumbelliferyl oleate, and tween 20 were obtained from Sigma-Aldrich Inc. (Beijing, China). Phosphate buffer, Tris-HCL buffer, citric acid buffer, ethylene glycol tetraacetic acid (EGTA), 2,4-dinitrophenylhydrazine, and Triton X-100 were produced by Thermo Technology Co., Ltd. (Beijing, China).

### 2.2. Hairtail Samples and Treatments

The air-dried hairtail were processed in a local factory in Zhoushan, Chian. Briefly, the gills, tail, viscera, and black membrane in the abdominal cavity of the hairtail samples were manually removed (but not sectioned). Next, the samples were coated with 2% (*w*/*w*) NaCl of its body weight on a sunny day, after which it was left to dry naturally under the sun for 2–5 days in outdoor natural conditions (wind speed: 2–6 m/s, relative humidity: 40–60%, and temperature: 4–7 °C).

The obtained air-dried hairtail with varying water contents, an average body length of 71 ± 4 cm, and an average weight of 223 ± 15 g were commercially obtained in September 2022 from a local aquatic market in Zhoushan (Zhejiang Province, China). The hairtail samples were divided into two groups depending on the water content, namely air-dried hairtail with a high water content of 71.8 ± 0.22% (abbreviated as HMAH) and air-dried hairtail a low water content of 62.6 ± 0.32% (abbreviated as LMAH). The acquired samples were placed on ice with a fish/ice ratio of 1:2 (*w*/*w*) in a cooler and transported to the Department of Food and Pharmacy, Zhejiang Ocean University. Upon arrival, the air-dried hairtail were sealed in a polyethylene bag, uniformly distributed, and maintained at 4 °C for 8 days (accelerated experimental conditions). During chilled storage, at different periods (0, 2, 4, 6, and 8 days), three packs of hairtail cuts with high and low water contents were randomly taken and used for the following analyses.

### 2.3. Water Activity, Water Content, and NaCl Content Analysis

The water activity, water content, and NaCl content of the air-dried hairtail samples were measured during chilled storage by using the ISO-recommended methods, 24557 [27], 1442 [28], and 1841-1 [29], respectively.

### 2.4. Lipid Oxidation Analysis

Crude lipid in the hairtail samples was extracted according to the method described by Pawliszyn [30], with some modifications. Briefly, 16 g of the samples was packed in a three-layer non-woven bag and placed in a thimble, and 70 mL of petroleum ether with a boiling point of 30–60 °C was added. Each sample was extracted for 5 h at a constant temperature of 55 °C. During this time, the crude lipid was collected with petroleum ether in a round-bottom flask using a rotary evaporator (RV 10 Auto, IKA, Baden-Wurttemberg, Germany). Finally, the extracted lipid was transferred into a 10 mL glass centrifuge tube and kept in a −80 °C freezer (905-ULTS, Thermo Fisher Scientific, Shanghai, China) except during experimental monitoring.

#### 2.4.1. Peroxide Value (PV) Analysis

The PV of the lipid extract was determined according to the Chinese National Standard, GB 5009.227 [31], with slight modifications. Briefly, the extracted lipid samples were dissolved in chloroform and glacial acetic acid solutions. The peroxides in the samples reacted with potassium iodide to produce iodine, which was then titrated with a sodium thiosulfate standard solution. The PVs were calculated as follows:PV (mmol/L) = [(V_1_ − V_0_) × c]/(2 × V) × 1000,
where V_1_ and V_0_ represent the volume of sodium thiosulfate solution consumed by the lipid samples and the reagent blank; c is the concentration of the sodium thiosulfate standard solution; and V is the volume of the lipid samples.

#### 2.4.2. Para-Anisidine Value (p-AnV) Analysis

The P-AnV of the lipid extract was determined in accordance with the Chinese National Standard, GB/T 24304 [32], and the report by Nan et al. [33]. Briefly, 0.2 g of the lipid extract was dissolved in 2.3 mL of isooctane. Next, the mixture was added to precisely 0.2 mL of an acetic acid solution and a p-anisidine reagent (2.5 g/L), which was further kept in the dark for 8 min at room temperature. The isooctane solvent was used as the blank. The absorbance was determined at a wavelength of 350 nm. The p-AnV was calculated as follows:p-AnV = 100 × Q × V/m × [1.2 × (A1 − A2 − A3)],
where Q (g/mL) is the concentration of the lipid sample; V (mL) is the volume of the lipid sample; and m (g) is the mass of the lipid sample. A1, A2, and A3 represent the absorbance at 350 nm of the lipid sample solution with p-anisidine added, the untested lipid sample solution with an equal amount of acetic acid added, and the control with p-anisidine added, respectively.

### 2.5. Non-Esterified Fatty Acid (NEFA) Content Analysis

The measurement of the NEFA content in the hairtail muscle was performed by using a commercial non-esterified free fatty acid assay kit supplied by the Nanjing Jiancheng Bioengineering Institute (Nanjing, China). The NEFA content was expressed as μmol/L of lipids.

### 2.6. Thiobarbituric Acid-Reactive Substance (TBARS) Value Analysis

The TBARS value of the hairtail muscle was determined according to the Chinese National Standard GB 5009.181 [34]. Briefly, 5 g of the fish samples was placed into a 100 mL stoppered conical flask, with 50 mL of TCA (70 g/L, 1 g/L EDTA-2Na) added. The flask was shaken, sealed, and oscillated at 50 °C for 30 min. Once cooled, the mixture was filtered. Then, 5 mL of the filtrate and standard series solutions were transferred to 25 mL colorimetric tubes. TCA was used as a blank, and 5 mL of TBA (0.02 mol/L) was added to each tube. The tubes were sealed, mixed, and placed in a 90 °C water bath for 30 min. After cooling, the absorbance of 3 mL of the supernatant and malondialdehyde (MDA) standard series at a wavelength of 532 nm was measured. The concentration of MDA in the samples was calculated from the standard curve drawn based on the MDA standard series. The TBARS (mg MDA eq/kg muscle) value was calculated as follows:TBARS = (C × V × 1000)/(M × 1000),
where C (μg/mL) is the concentration of MDA calculated from the standard curve; V (mL) represents the volume of the resulting solution; and M (g) is the mass of the sampled muscle.

### 2.7. Lipase Enzyme Activity Analysis

#### 2.7.1. Crude Lipase Enzyme Extraction

The crude lipase enzyme was extracted according to the modified Hernandez method [35]. Briefly, 5 g of the hairtail samples was accurately weighed and added with 25 mL of phosphate buffer (50 mmol/L, pH 7.5) containing 5 mmol/L of EGTA. After being homogenized at 15,000 r/min for 6 × 10 s, the mixture was uniformly stirred in an ice bath for 30 min and then centrifuged at 4 °C and 10,000× *g*. The resulting supernatant was filtered with four layers of gauze. Afterward, the crude lipase enzyme was fixed with 25 mL of phosphate buffer. 

#### 2.7.2. Neutral Lipase Activity Analysis

The neutral lipase activity was determined according to Motilva’s method [36], with some modifications. Briefly, 0.1 mL of crude lipase extract was added into 2.8 mL of Tris-HCl buffer (220 mmol/L, pH 7.5) containing 0.05% triton x-100. Next, the mixture was reacted with 0.1 mL of 4-methylumbelliferyl oleate (1 mmol/L) for 30 min at 37 °C, which was stopped by the addition of 2 mL of anhydrous ethanol. The mixture was measured at excitation and emission wavelengths of 328 and 470 nm, respectively, in a fluorospectrophotometer (F97Pro, Lengguang Technology, Shanghai, China). One enzymatic unit was defined as an increase in absorbance of 1 per minute per gram of the air-dried hairtail muscle at 328 nm and 37 °C. The neutral lipase activity (U/g) was calculated as follows:Neutral lipase activity = (A1 − A0) × V0/(M × V2/V1)/T,
where A0 and A1 are the absorbances at 328 nm and 470 nm, respectively; V0 (mL) represents the total volume of the reaction system; M (g) is the mass of the sample; V2 is the volume of the crude enzyme supernatant added to the reaction system; V0 is the total volume of the crude enzyme supernatant; and T (min) represents the reaction time.

#### 2.7.3. Alkaline Lipase Activity Analysis

The alkaline lipase activity was determined according to Bicas’s method [37], with slight modifications. Briefly, 161 mg of p-NPL was dissolved in 80 mL of acetate buffer (50 mmol/L, pH 5.6) containing 1% triton x-100. Next, 5 mL of this substrate was reacted with 0.5 mL of the crude enzymatic extract in 4.5 mL of Tris-HCl buffer (50 mmol/L, pH 8.0). After exactly 10 min at 55 °C, the reaction was stopped by the addition of 10 mL of anhydrous ethanol. The absorbance of the supernatant at 410 nm was measured. One enzymatic unit was defined as an increase in absorbance of 1 per minute per gram of the air-dried hairtail muscle at 410 nm and 55 °C. The alkaline lipase activity (U/g) was calculated using the same formula as in Section 2.7.2, where A1 and A0 represent the absorbance at 410 nm of the test group and control group, respectively. The control group was reacted with 5 mL of acetate buffer (50 mmol/L, pH 5.6) containing 1% triton x-100 without p-NPL.

### 2.8. Lipoxygenase Activity Analysis

The crude lipoxygenase extraction and its activity were measured according to the modified Gata method [38]. Briefly, 5 g of the air-dried hairtail samples was accurately weighed and added to 20 mL of phosphate buffer (50 mmol/L, pH 7.0, containing 1 mmol/L of DTT and 1 mmol/L of EGTA). After homogenizing at 15,000 r/min for 6 × 10 s, the samples were uniformly stirred in an ice bath for 30 min and then centrifuged at 10,000× *g* at 4 °C for 10 min. The supernatant was filtered with four layers of gauze and fixed with 25 mL of phosphate buffer to prepare the crude enzyme extraction.

The lipoxygenase activity was determined at room temperature according to the modified Bian method [39]. Briefly, 0.2 mL of linoleic acid in double-distilled water (containing 10 mol/L of linoleic acid and 3.6 mL/L of tween 20) was mixed with 2.7 mL of citric acid buffer (50 mmol/L, pH 5.5), and the absorbance increased at 234 nm for 1 min before and after adding 0.1 mL of the crude enzyme extraction. One unit of lipoxygenase activity was defined as an increase in absorbance of 1 per minute per gram of the air-dried hairtail muscle at 234 nm and 25 °C. The lipoxygenase activity (U/g) was calculated using the same formula as in Section 2.7.2, where A1 and A0 represent the absorbance at 234 nm of the test group and control group, respectively.

### 2.9. Data Analysis

All experiments were performed in triplicate. All results presented were analyzed using a t-test or one-way analysis of variance using SPSS 24.0 (SPSS-Statistical Package for the Social Sciences, IBM, New York, NY, USA) and presented as mean ± standard deviation, which was used to identify significant differences (*p* < 0.05).

## 3. Results and Discussion

### 3.1. Water Activity, Water Content, and NaCl Content Analysis

Figure 1 illustrates the changes in the water activity, water content, and NaCl content of the air-dried hairtail samples stored at 4 °C for 8 days. The alterations in water content (Figure 1A) and water activity (Figure 1B) of the HMAH and LMAH showed similar trends with the duration of chilled storage. At the initial stage of storage, the water activity and water content of the HMAH were 0.999 and 71.82%, respectively. These surpassed those of the LMAH, which were 0.779 and 62.64%, respectively. During the following storage, the water content of the HMAH was reduced due to low external humidity. Conversely, the LMAH absorbed the water molecules from the environment, resulting in an increased water content and water activity. Over time, both the HMAH and LMAH exhibited a tendency to equilibrate with the inherent humidity of the extrinsic environment. The change in water content and water activity further affected the NaCl content in the hairtail samples, exhibiting an inverse trend compared with those of the NaCl content (Figure 1C) of both the HMAH and LMAH.

### 3.2. Primary Oxidation Analysis

The results of the primary oxidation (CD and PV) of the HMAH and LMAH are depicted in Figure 2, which exhibit upward followed by downward trends throughout the 8 days of chilled storage. Initially, hydrogen peroxide and several metabolic intermediates were formed and accumulated due to the primary oxidation. The higher NaCl content might be an important contributing factor to the accelerated primary lipid oxidation rate in the LMAH samples compared to that in the HMAH samples, thereby reaching the maximum value more rapidly. Previous research revealed a certain interval wherein the NaCl content was less than 1%; increasing the NaCl content expedited the primary oxidation process during dry-salted bacon processing [24]. The CD (Figure 2A) and PV (Figure 2B) of the LMAH and HMAH commenced downward trends at 4–6 d, respectively, where the consumption of PV and CD in the secondary oxidation reaction might surpass their production. The maximum PV and CD in the LMAH samples were comparatively lower than those of the HMAH samples, possibly suggesting that a lack of water inhibited lipid oxidation during the primary oxidation. Vu et al. [25] also reached similar conclusions; the low water activity effectively suppressed the accumulation of peroxides in low-moisture foods, particularly at water activity levels ranging from 0.5 to 1, where the inhibitory effect was more noticeable.

### 3.3. Secondary Oxidation Analysis

The secondary oxidation of lipids in the air-dried hairtail samples was quantified based on an evaluation of the TBARS value and p-AnV during refrigeration at 4 °C (Figure 3). The TBARS value (Figure 3A) and p-AnV (Figure 3B) of the HMAH and LMAH samples showed analogous and arresting upward trends throughout preservation. This further confirmed the inhibitory effect of low water contents on lipid secondary oxidation. Previous investigations also found an increasing trend of TBARS in air-dried hairtail and fresh hairtail samples during storage at 4 °C. Nonetheless, the TBARS in the air-dried hairtail significantly exceeded that of the fresh hairtail during chilled storage at 4 °C, constituting an unavoidable outcome following the salination and air-drying process [4]. Nevertheless, the trend of TBARS of air-dried hairtail was similar to that of fresh hairtail [40]. Under experimental conditions, while preparing air-dried hairtail (2% NaCl), lipid secondary oxidation was present right from the commencement of salination [5]. At the initial stage of chilled storage, the content of secondary oxidation products of the HMAH samples was marginally higher than that of the LMAH samples. At 0 days, the TBARS value and p-AnV of the HMAH samples were determined as 15.36 mg of MDA/g of muscle and 1.68, respectively, which were slightly higher than those of the LMAH samples (14.88 mg of MDA/g of muscle and 1.51, respectively). This indicated an overall similarity in the initial secondary oxidation levels between the LMAH and HMAH samples. Micromolecules such as aldehydes, ketones, and quinones were regarded as the typical biomarkers depicting lipid secondary oxidation at the initial stage of storage. Moreover, the progress of the secondary oxidation in both samples was delayed during the initial storage, which indirectly induced the rapid accumulation of primary oxidation products. During the later stage of storage, the secondary oxidation of lipids experienced a substantial increase. Simultaneously, the consumption of hydroperoxide, free radicals, reactive oxygen, and CD outpaced the generation rates, ultimately inducing a downward trend in the PV and CD levels. Previous research also demonstrated a similar uptrend of the p-AnV and TBARS value in mackerel (*Scomber japonicus*) that was pre-soaked in a k-carrageenan oligosaccharide bath during various stages of frozen storage [41].

### 3.4. Neutral Lipase, Alkaline Lipase, and Lipoxygenase Activity Analysis

Changes in the activities of neutral and alkaline lipases, as well as lipoxygenase, in the air-dried hairtail samples are shown in Figure 4. The activities of both neutral lipase (Figure 4A) and alkaline lipase (Figure 4B) in the HMAH and LMAH samples increased with the progression of chilled storage. The growth rate of the neutral lipase activity of the HMAH and LMAH samples reached the maximum levels at 4 d. The alkaline lipase activity of the HMAH samples exhibited a rapid uptrend at 0–4 d, and the enzyme activity’s growth rate declined after 4 d of storage. During the initial stage of storage, the neutral and alkaline lipase activities of the LMAH samples were similar to those of the HMAH samples. Moreover, the growth rates for both neutral and alkaline activities in the LMAH samples were comparatively lower than those in the HMAH samples. The lipoxygenase activity (Figure 4C) of the air-dried hairtail samples significantly declined as the chilled storage time increased. Additionally, the lipoxygenase activity of the LMAH samples consistently remained at lower levels than that of the HMAH samples during the entire storage. The reduction in water content might inhibit the activity of neutral and alkaline lipases, as well as lipoxygenase. Previous research indicated that the enzymatic hydrolysis of lipids was significantly promoted by increases in water activity in food models [42]. A similar reduction in lipoxygenase activity in fresh Russian sturgeon (*Acipenser gueldenstaedti*) steak was observed during chilled storage at 4 °C [43]. In contrast, the lipoxygenase activity of grass carp (*Ctenopharyngodon idella*) that was pre-treated with 0.3% chlorogenic acid showed notable increases during short-term preservation at 4 °C, and then showed a downward trend after 8 days [44]. In addition, previous studies also suggested that the neutral or alkaline biocatalysis of lipids might be triggered by other enzymes present in the adipose tissue [45].

### 3.5. NEFA Content Analysis

The NEFA content of the HMAH and LMAH samples increased over the period of chilled storage, as shown in Figure 5. Additionally, the NEFA content of the LMAH samples was lower than that of the HMAH samples, presumably due to the inhibition of lipid hydrolysis by the lower water content in their muscle tissues. To metabolize the NEFAs, the lipid hydrolysis required water molecules as a reagent to break the ester bonds in lipids. Previous reports indicated that the rate of lipid hydrolysis was heavily influenced by the availability of water molecules in the reaction system [20]. The observations in the NEFA contents influenced by water content were similar to those reported in [46] in the investigation on salted herring (*Clupea harengus*) products.

### 3.6. Correlation Analysis

The results of the subsequent correlation analysis (Figure 6) indicated that several indexes evinced significant positive or negative correlations (*p* ≤ 0.05). Each index of the HMAH (Figure 6A) and the LMAH (Figure 6B) samples, including lipid hydrolysis, secondary lipid oxidation, lipase activity, and lipoxygenase activity, individually demonstrated positive or negative correlations. Both the low water activity and high NaCl content inhibited the growth rates of the aforementioned indicators in the LMAH samples, further explaining the low water content itself, along with the resulting high NaCl content environment, which significantly inhibited lipid oxidation in the hairtail samples.

Both the TBARS value and p-AnV in the HMAH and LMAH samples showed a significant positive correlation (*p* ≤ 0.05), which described lipid secondary oxidation. MDA was one of the many products of lipid hydroperoxides in the secondary oxidation, which was easy to accumulate when metabolism stopped [9]. Meanwhile, the hydroperoxide is broken down into a short-chain aldehyde material, developing a rancid flavor in lipids [47]. This further confirmed that TBARS and p-AnV measured different oxidation products of lipids, which may still be accumulated simultaneously as the secondary oxidation progresses.

The neutral lipase activity, alkaline lipase activity, and NEFA content in the HMAH and LMAH samples all demonstrated significant positive correlations with each other (*p* ≤ 0.05), except that the positive correlation between the NEFA content and neutral lipase activity in the LMAH samples was not significant (*p* ≥ 0.05). This shows that lipase activity was the main factor in lipid hydrolysis. Although there was spontaneous lipid hydrolysis, lipases were essential to accelerate the reaction rate [48,49].

As lipids were metabolized through oxidation and hydrolysis in the fish during storage, triglyceride molecules were cleaved to release the free fatty acids. These free fatty acids mainly contributed to the increased NEFA content. Hormone-sensitive lipase and monoglyceride lipase were the key hydrolases during the degradation of triglycerides and diglycerides, and they played important roles during the hydrolysis of the monoglyceride into adipocytes [50]. Two hormone-sensitive lipases were found in rainbow trout (*Oncorhynchus mykiss*), suggesting that distinct mechanisms serve to regulate the differential expression of these two hormone-sensitive lipases [51]. The 1,3-specific lipases performed the task of decomposing triacylglycerols at positions Ci and C3, which produce fatty acids and 2-monoacylglycerols, alongside 1,3 or 2,3 diacylglycerols [52]. Together, these lipids decomposed the triglycerides into diglycerides, and further into monoglycerides and free fatty acids.

The neutral and alkaline lipase activities in the HMAH samples showed a significant positive correlation with the NaCl content (*p* ≤ 0.05), while the neutral and alkaline lipase activities in the LMAH samples showed a significant negative correlation with the NaCl content. This phenomenon suggests that the neutral and alkaline lipase activities in the air-dried hairtail samples may also be influenced by the NaCl content during chilled storage. Previous research provided a similar suggestion; that is, NaCl content within an appropriate range reduced the chemical energy required for lipid oxidation. The accumulation of peroxide products in salted bacon during storage was inhibited as the NaCl content increased from 1% to 4% [24].

Lipoxygenase activity showed a negative correlation with both primary and secondary lipid oxidation, which indicated that the negative growth of the lipoxygenase activity did not effectively inhibit further lipid oxidation. This indicates that lipoxygenase was not a key factor during hairtail lipid oxidation. Previous studies showed that the type and mechanism of oxidation of the substrate by lipoxygenase were selective. Fatty chains with cis, cis-1,4-pentadiene moiety in lipids were a prerequisite for lipoxygenase substrates [53]. In addition to fatty acids, some glycerides, fatty esters, and other fatty derivatives could also serve as reaction substrates for some homologous families of lipoxygenase [54].

The NEFA content had a significant positive correlation with the TBARS value and p-AnV in both the HMAH and LMAH samples (*p* ≤ 0.05). The secondary oxidation reaction of lipids was inactive in the early stage of air-dried hairtail storage. The increase in the NEFA content in the air-dried hairtail was primarily due to the free fatty acids produced by lipid hydrolysis. Affected by the secondary oxidation of lipids, the total amount of hydrogen peroxides in the HMAH and LMAH samples began to decrease significantly from 4 days and 6 days, respectively. Short-chain fatty acids of hairtail samples during a later stage of storage were further induced to increase the NEFA content through the decomposition of peroxide and the oxidation of aldehydes. Therefore, the NEFA content simultaneously evaluated the level of lipid hydrolysis and oxidation in the air-dried hairtail samples.

The NEFA content has been extensively contemplated to be associated with lipid oxidation describing lipid acid spoilage. This was primarily due to the decomposition of peroxides that directly released hydrogen-ion lactones, as well as secondary oxidation products such as acids, aldehydes, ketones, and alkenes. These secondary oxidation products could be further oxidized into carboxylic acids under the influence of free radicals or reactive oxygen [21,23]. A high-temperature lipid oxidation treatment was proven to be the key factor for the significant increase in NEFA content in refined grapeseed oil and refined olive oil [55]. The observation of the correlation between the NEFA content and lipid oxidation was similar to that reported by Vázquez et al. [56] in their research on frozen hake (*Merluccius merluccius*).

## 4. Conclusions

The physicochemical quality of air-dried hairtail was studied for a duration of 8 days of chilled storage at 4 °C. This study revealed the variations in the physicochemical indices of two different initial water contents of the air-dried hairtails, which deteriorated over time during chilled storage. Notably, the low-water-content air-dried hairtails presented a substantially lower instance of lipid deterioration compared to the high-water-content air-dried hairtail samples. This phenomenon is demonstrated by lower levels of primary and secondary oxidation of lipids, as well as the suppression of the activity of lipase and lipoxygenase. By controlling the initial water content of air-dried hairtails during processing and storage, it is possible to better retain their nutritional value and flavor, additionally extending their shelf-life.

## Figures and Tables

**Figure 1 foods-13-00229-f001:**
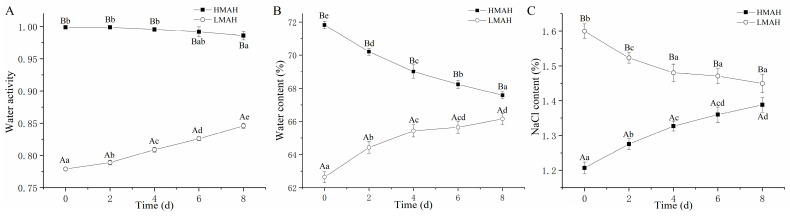
Changes in the water activity (**A**), water content (**B**), and NaCl content (C) of high-water-content air-dried hairtail (HMAH) and low-water-content air-dried hairtail (LMAH) samples during 8 days of chilled storage. Different lowercase letters for the same group during 8 days of storage indicate significant differences (*p* < 0.05), and different uppercase letters for the two groups at the same storage time indicate significant differences (*p* < 0.05).

**Figure 2 foods-13-00229-f002:**
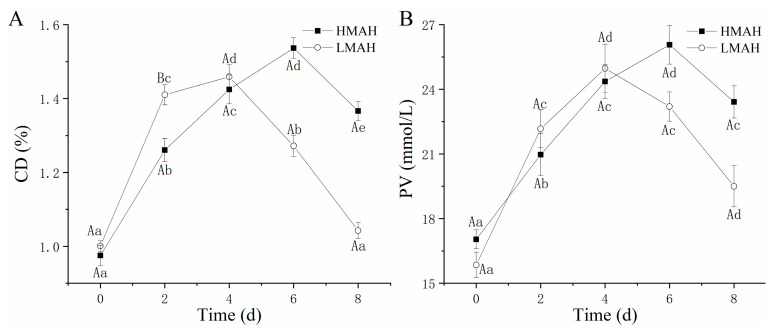
Changes in the CD (**A**) and PV (**B**) of high-water-content air-dried hairtail (HMAH) and low-water-content air-dried hairtail (LMAH) samples during 8 days of chilled storage. Different lowercase letters for the same group during 8 days of storage indicate significant differences (*p* < 0.05), and different uppercase letters for the two groups at the same storage time indicate significant differences (*p* < 0.05).

**Figure 3 foods-13-00229-f003:**
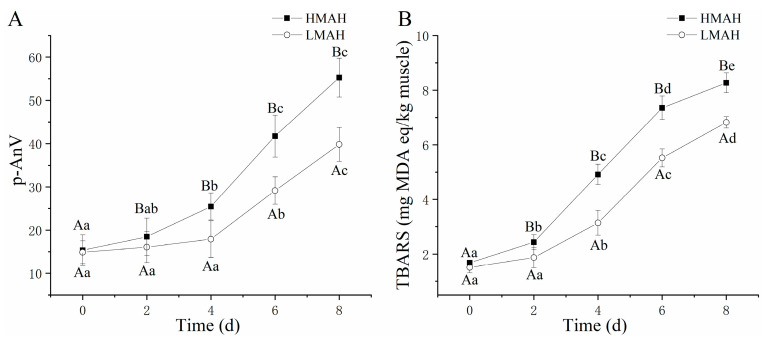
Changes in the p-AnV (**A**) and TBARS value (**B**) of high-water-content air-dried hairtail (HMAH) and low-water-content air-dried hairtail (LMAH) samples during 8 days of chilled storage. Different lowercase letters for the same group during 8 days of storage indicate significant differences (*p* < 0.05), and different uppercase letters for the two groups at the same storage time indicate significant differences (*p* < 0.05).

**Figure 4 foods-13-00229-f004:**
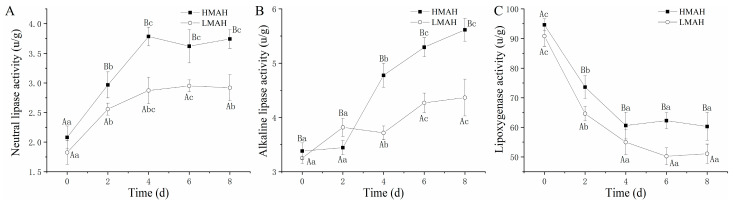
Changes in neutral lipase activity (**A**), alkaline lipase activity (**B**), and lipoxygenase activity (**C**) of high-water-content air-dried hairtail (HMAH) and low-water-content air-dried hairtail (LMAH) samples during 8 days of chilled storage. Different lowercase letters for the same group during 8 days of storage indicate significant differences (*p* < 0.05), and different uppercase letters for the two groups at the same storage time indicate significant differences (*p* < 0.05).

**Figure 5 foods-13-00229-f005:**
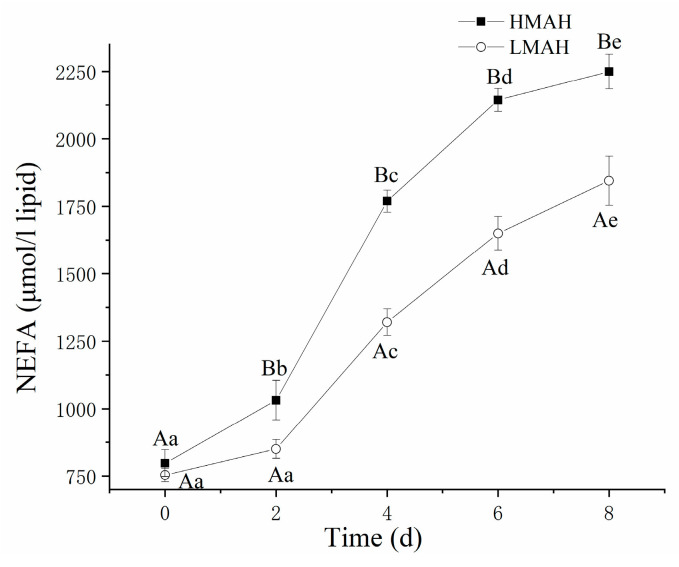
Changes in the NEFA content of high-water-content air-dried hairtail (HMAH) and low-water-content air-dried hairtail (LMAH) samples during 8 days of chilled storage. Different lowercase letters for the same group during 8 days of storage indicate significant differences (*p* < 0.05), and different uppercase letters for the two groups at the same storage time indicate significant differences (*p* < 0.05).

**Figure 6 foods-13-00229-f006:**
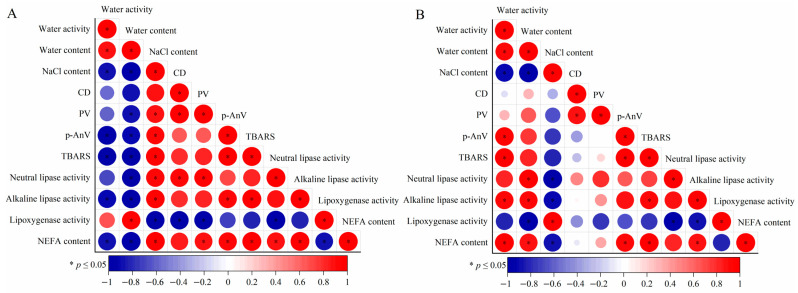
Correlation analysis of the physicochemical parameters in the high-water-content air-dried hairtail (HMAH) (**A**) and low-water-content air-dried hairtail (LMAH) (**B**) samples. The blue and red dots represent the negative and positive correlations between the two indices, respectively. The asterisk symbol inside the red plot indicates a significant positive correlation, and that inside the blue plot indicates a significant negative correlation.

## Data Availability

Data is contained within the article.

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
