# Peer review of "Investigation of the Alternations in Lipid Oxidation and Lipase Activity in Air-Dried Hairtail (Trichiurus lepturus) during Chilled Storage"

_foods, 2024, doi:10.3390/foods13020229_

Round 1

Reviewer 1 Report

Comments and Suggestions for Authors

The article in general, is well written.

Line 210 - Please change "Results" to "Results and Discussion," as the authors did not separate the section.

What is the difference between neutral lipase and alkaline lipase? Why did the authors decide to measure these? 

Regarding statistical data, please add the statistical information on the results. As presented,  All Figures are without any information on the analysis of statistical results, we cannot tell the difference of sample during storage or between HMAH and LMAH. Please revise the data presentation and ensure it is written on the results and discussion.

For comparison of results found in this study with the literature, it is suggested that the authors should cite more related to hairtail oxidation, as it is a more suitable citation in this study. Please replace them with more related citations (Line 276, Line 302, Line 403). 

Reviewer 2 Report

Comments and Suggestions for Authors

This study investigated the lipid oxidation of dried hairtail during storage. While this manuscript provides some theoretical support, the overall experimental design is nothing new but a repetition of previous works. In particular, the study design is vague, and no explanation was provided about drying. My other comments are:

Line 14, which activity?

Line 16, One should mention how they were dried and why NaCl was added before the results. 

Lines 18-19, value is not a part of abbreviation? 

Line 36, One should include the % of PUFAs

Line 38, if it is naturally dried, it should be labeled as "sun drying," as "air drying" is often referred to as hot air drying. 

Line 45, "intermediate products"

Lines 45-46, is this a correct statement? 

Line 65, "activity. In the current study"

Section 2.2, how were these samples dried? How were the environmental conditions like temperature, humidity, wind speed, etc.?

Line 90, why was it stored for only 8 days? What's the overall shelf life of these materials? 

Line 94, where did this NaCl come from?

In Figure 2, the CD and PV values start decreasing after 4 days, so a scientific explanation should be provided. Authors may follow this article (https://doi.org/10.1016/j.foodres.2022.1122620), where the reduction of TBARS during storage was explained. 

Figure 3, "(mg MDA eq/ kg)"

Comments on the Quality of English Language

Minor editing of English language required

Round 2

Reviewer 2 Report

Comments and Suggestions for Authors

The authors have now responded to the questions accordingly, and it is ready to proceed to the next steps.

Comments on the Quality of English Language

The authors have now responded to the questions accordingly, and it is ready to proceed to the next steps.